

# Multiplex *vs.* singleplex assay for the simultaneous identification of the three components of avian malaria vector-borne disease by DNA metabarcoding

Eduard Mas-Carrió[1], Judith Schneider[1], Victor Othenin-Girard[1], Romain Pigeault[2,3], Pierre Taberlet[4,5], Philippe Christe[2], Olivier Glaizot[2,6] and Luca Fumagalli[1,7]

[1] Laboratory for Conservation Biology, Department of Ecology and Evolution, Biophore, University of Lausanne, Lausanne, Switzerland
[2] Department of Ecology and Evolution, Biophore, University of Lausanne, Lausanne, Switzerland
[3] Laboratoire EBI, Equipe EES, UMR CNRS 7267, Université de Poitiers, Poitiers, France
[4] CNRS, Laboratoire d'Ecologie Alpine, Université Grenoble Alpes, Grenoble, France
[5] UiT, Tromsø Museum, The Arctic University of Norway, Tromsø, Norway
[6] Department of Zoology, State Museum of Natural History, Lausanne, Switzerland
[7] Swiss Human Institute of Forensic Taphonomy, University Centre of Legal Medicine Lausanne-Geneva, Lausanne University Hospital and University of Lausanne, University of Lausanne, Lausanne, Switzerland

Corresponding author
Luca Fumagalli,
luca.fumagalli@unil.ch

## ABSTRACT

Accurate detection and identification of vector-host-parasite systems are key to understanding their evolutionary dynamics and to design effective disease prevention strategies. Traditionally, microscopical and serological techniques were employed to analyse arthropod blood meals for host/parasite detection, but these were limited in taxonomic resolution and only to pre-selected taxa. In recent years, molecular techniques have emerged as a promising alternative, offering enhanced resolution and taxonomic range. While singleplex polymerase chain reaction (PCR) assays were used at first to identify host, vector and parasite components in separate reactions, today multiple primer pairs can be combined in a single reaction, *i.e.*, multiplex, offering substantial time and cost savings. Nonetheless, despite the potential benefits of multiplex PCR, studies quantifying its efficacy compared to singleplex reactions are scarce. In this study, we used partially digested mosquito blood meals within an avian malaria framework to jointly identify the host, vector and parasite using multiplex DNA metabarcoding, and to compare it with separate singleplex PCRs. We aimed to compare the detection probabilities and taxonomic assignments between both approaches. We found both to have similar performances in terms of detection for the host and the vector, but singleplex clearly outperformed multiplex for the parasite component. We suggest adjusting the relative concentrations of the PCR primers used in the multiplex assay could increase the efficiency of multiplex in detecting all the components of the studied multi-species system. Overall, the results show that multiplex DNA metabarcoding can be an effective approach that could be applied to any vector-borne interaction involving blood-feeding arthropods. Our insights from

this proof-of-concept study will help improve laboratory procedures for accurate and cost-efficient medical diagnosis of vector-borne diseases, the spread of which is globally exacerbated by current climate change.

## INTRODUCTION

Accurate detection and identification of vector-host-parasite interactions is crucial for understanding the complex evolutionary dynamics of these systems and for managing prevention campaigns (*Almeida et al., 2024*; *Casman & Dowlatabadi, 2010*; *Martinez et al., 2021*). Different methodological approaches have been used to analyse this tripartite interaction, traditionally focusing on arthropod blood meals, using serological techniques such as enzyme-linked immunosorbent assay or precipitin tests (*e.g., Beier et al., 1988*; *Tempelis, 1975*; *Washino & Tempelis, 1983*). However, in these studies taxonomic resolution was low (*Lardeux et al., 2007*) and the detection was limited to pre-selected taxa. Over the past twenty years, blood meals (or digestive parts) of various blood-feeding invertebrate taxa have been analysed with molecular techniques (*i.e.,* polymerase chain reaction, PCR) to better characterise these interactions (reviewed in *Borland & Kading, 2021*). This has been specifically conducted for the identification of either parasites (*e.g., Ciloglu et al., 2019*; *Szentiványi et al., 2020*), hosts (*e.g.,* fed on by leeches (*Abrams et al., 2019*; *Schnell et al., 2018*; *Weiskopf et al., 2018*)), triatomines (*Balasubramanian et al., 2022*), midges (*Cutajar & Rowley, 2020*), sandflies (*e.g., Kocher et al., 2017*) or mosquitoes (*e.g., Estrada-Franco et al., 2020*), hosts and parasites (*e.g., Dumonteil et al., 2018*) or vectors and parasites (*Makunin et al., 2022*). Often based on Sanger-sequencing, these approaches fail to distinguish mixed blood meals or parasite co-infections, and require a phasing step (or a nested PCR protocol). Furthermore, most of these assays target relatively large-size DNA barcodes (>250 bp; *e.g., Alcaide et al., 2009*), preventing the successful amplification of degraded samples and/or with very low amount or degraded DNA of the target species.

Until recently, singleplex PCR assays (the use of a single primer pair per PCR reaction) were the only way to detect the components of a vector-borne system, *i.e.,* by performing separate PCR reactions for the host, vector and parasite. Some studies have already used singleplex to identify the host-vector-parasite triad, *e.g.,* using blackflies (*Chakarov et al., 2020*) or triatomines (*Dumonteil et al., 2018*).

Nowadays, advances in primer design, next-generation sequencing and the use of DNA metabarcoding approaches have opened the possibility for identifying vector-borne interactions in a single PCR reaction combining multiple primer pairs, *i.e.,* multiplex PCR, which greatly reduces the laboratory costs and time. Currently, the application of multiplexing is however limited, as there is a general lack of multiplexing routine among the community and of its potential for downscaling costs. However, the use of *in silico* PCR (*i.e.,* computational tools to simulate PCR amplification) has simplified the application of

multiplex, facilitating the design of new primer pairs to be both taxonomically informative and compatible in terms of PCR reaction conditions. *A priori*, singleplex PCR is easier to design and is expected to result in higher detection scores of the target species, because only one target is amplified per reaction without potential primer competition during the PCR reaction, and conditions are optimised for only one primer pair. When performing multiplex PCRs, the initial concentration of DNA from each target taxonomic group may compromise the successful amplification with primer pairs aimed at less abundant taxa. This could occur either due to the consumption of PCR reagents by one or a few primers, or because disproportionate amplification by one primer pair saturates the sequencing clusters.

In this proof-of-concept study, we used short DNA metabarcodes for the amplification and high-throughput sequencing of partially digested mosquito blood meals in an avian malaria framework as a case study system. This vector-borne disease is caused by apicomplexan parasites belonging to the *Plasmodium* genus, which are transmitted in birds by mosquitoes. For the first time, mosquito blood meals were used here to simultaneously identify, using multiplex DNA metabarcoding, the host-vector-parasite triad in a tripartite vector-borne system.

In order to improve the identification methodology, we compared the detection probabilities by singleplex and multiplex DNA metabarcoding of the three components of this system and the differences in their taxonomic assignments. Studies quantifying the detection differences between singleplex and multiplex reactions are still rare (*Eastwood et al., 2024*). As such, expanding our knowledge on which factors determine the successful detection of the components of vector-host-parasite interactions comparing singleplex and multiplex approaches will help to develop more cost-efficient laboratory procedures and improve medical diagnosis of vector-borne diseases, whose transmission and dispersion into non-endemic areas are exacerbated by current climate change (*Thomson & Stanberry, 2022*).

## MATERIALS & METHODS

### Sample collection

Thirty-six wild visually-identified blood-fed female mosquitoes were captured between June and September 2019 in two distinct locations in Switzerland (Dorigny (46°31′N, 6°34′E) and Monods (46°34′N, 6°24′E) forests, Canton Vaud; see Data S1 material for the full metadata details). Field experiments were approved by the Direction générale de l'environnement (DGE), Section biodiversité et paysage, CH-1014 Lausanne (2019-3978). In addition, 15 *Culex pipiens* mosquitoes reared in the laboratory and fed on experimentally *Plasmodium*-infected captive birds (*Serinus canaria*, Fringillidae, and *Passer domesticus*, Passeridae) were included as positive controls. The abdomens of all 51 mosquitoes were dissected under a binocular microscope using sterilised Vannas-Tübingen Spring Scissor (no. 15003-08; Fine Science Tools) and stored at −80 °C until DNA extraction.

## DNA extraction

Prior to DNA extraction, mosquito abdomens were grinded with homogenisation beads on a tissue homogeniser (Bertin Technologies, Montigny-le-Bretonneux, France). DNA extraction was done using a DNeasy Blood & Tissue Kit (Qiagen, Hilden, Germany) following the manufacturers' protocol but with the elution step modified as follows: (i) 75 μL of elution buffer and 15 min of incubation at room temperature, followed by centrifugation, and (ii) 100 μL of elution buffer and one minute of incubation followed by centrifugation.

## Primer description and preliminary tests

Three primer pairs targeting different mitochondrial DNA (mtDNA) genes were used to identify the blood meal origin (*i.e.,* host), the mosquito species (*i.e.,* vector) and the parasite based on the DNA extracted from the 51 mosquito abdomens.

First, a metabarcode was amplified to identify the blood source using *Aves02* (*Taberlet et al., 2018*), targeting the 12S rDNA gene in birds. As *Aves02* primers were only tested using *in silico* PCR so far, preliminary PCRs with these primers were done on tissue DNA extracts (liver, muscle, blood) of 21 avian species from a wide range of orders (Table S1). These were verified on a 2% agarose gel. The results of this test showed that the targeted *Aves02* metabarcode was not amplified for the Eurasian blue tit (*Cyanistes caeruleus*), which is a common species in the study sites. Therefore, based on Passeriformes sequences complementary to *Aves02* primers found in GenBank, two new versions of the forward *Aves02* primer targeting the blue tit *Cyanistes caeruleus* (*Aves02_Cyan_F*) and the great tit *Parus major* (*Aves02_Parus_F*; also common in the study sites) were designed (Table S2) to better cover the avian taxonomic panel. These two modified versions were mixed, in a proportion of 25% each, with the standard forward *Aves02* primer for the final PCRs (the mixture of these three forward and the original reverse PCR primers is thereafter noted as *Aves02_mix*). In case of failed amplifications with *Aves02_mix* primers, samples were tested using a more generalist vertebrate primer (*Vert01*, *Riaz et al., 2011*; *Taberlet et al., 2018*) to check if the host was a non-avian vertebrate or to confirm the negative detection of the host in the blood. Second, to identify the mosquito vector, we used a primer pair targeting the 16S rDNA gene in Culicidae (*Culi01*; *Schneider et al., 2016*). Third, primers targeting the genus *Plasmodium* and amplifying a 37–44 bp portion of the cytochrome *b* gene were newly designed for this study (*Plasmo01*) using the *OBITools* package (*Boyer et al., 2016*). Since among Haemosporidian parasites, mosquitoes only transmit *Plasmodium*, we only targeted this genus. In brief: we used *ecoPrimers* software (*Ficetola et al., 2010*; *Riaz et al., 2011*) to find the best suitable primer for this purpose based on all complete mtDNA genomes available in GenBank. It was validated with a *in silico* PCR using *ecoPCR* software (*Ficetola et al., 2010*) for the targeted taxonomic group, using the EMBL database release 142 (January 2020). The results showed that *Plasmo01* covered 100% of the target group, with 12 mitogenomes amplified *in silico* available in GenBank, and that the taxonomic resolution was 57.1% at species level, and 100% at genus and family levels. Despite its short length, the barcode was retained because it provides the best taxonomic resolution across the *Plasmodium* genus given the used database. As *Plasmo01* primers were also only

tested using *in silico* PCR so far, preliminary PCRs were performed on DNA extracts of 18 *Plasmodium* mtDNA lineages (Table S1). The presence of DNA amplicons at the expected molecular size was verified on an agarose gel.

All blood meal samples were amplified in triplicate with the three primer pairs in both singleplex and multiplex assays.

## Singleplex DNA metabarcoding

To test for the presence of PCR inhibitors, preliminary PCRs were performed with each primer pair on eight randomly selected samples with different dilutions, and the amplicons band intensities checked on agarose gels. Based on these results, DNA extracts from wild blood-fed mosquitoes were diluted 2-fold and those from laboratory blood-fed mosquitoes 2,000-fold before being amplified with *Aves02_mix*. DNA extracts were diluted 10-fold before being amplified with *Culi01*, and no dilution was applied for the amplification with *Plasmo01*.

The PCR reagents and their final concentrations were the following: AmpliTaq Gold 360 Master Mix 1x, tagged forward and reverse primers at 0.5 µM, BSA at 0.16 mg/mL. The final volume per well was 20 µl including two µL of template DNA. The PCR thermal profile started with denaturation at 95 °C for 10 min, followed by 40 cycles with *Aves02_mix*, 45 cycles with *Plasmo01* and 50 cycles with *Culi01*. Each cycle was composed of 30 s at 95 °C, 30 s at 56 °C, 55 °C and 60 °C for *Aves02_mix*, for *Plasmo01* and for *Culi01* primers, respectively, and one minute at 72 °C, before a final elongation at 72 °C for seven minutes.

## Multiplex DNA metabarcoding

The three primer pairs used for singleplex were mixed to perform a single PCR reaction using the following final concentrations: *Aves02_mix* 0.4 µM, *Plasmo01* 0.4 µM, *Culi01* 0.15 µM. Final concentrations of the other PCR reagents were the same as in the singleplex assay, as well as final and template DNA volumes. We reduced the concentration of *Culi01* because the preliminary tests revealed an overamplification of mosquito sequences with equal primer concentration (data not shown). For the multiplex PCR reactions, all DNA extracts were diluted 10-fold. The PCR conditions were as follows: initial denaturation at 95 °C for 10 min, followed by 45 cycles with 30 s at 95 °C, 30 s at 55 °C and 1 min at 72 °C; final elongation at 72 °C for seven minutes.

## Purification, library preparation and sequencing

For each assay, extraction negative, PCR negative and positive controls (Table S3) as well as blanks were included in each PCR plate (for details on plate layout, see *Taberlet et al., 2018*). Before pooling amplicons per plate, amplification success was verified on an agarose gel for a subset of samples.

Amplicon pools were purified using the MinElute PCR Purification Kit (Qiagen, Hilden, Germany) and quantified using a Qubit 2.0 Fluorometer (Life Technology Corporation). Library preparation was performed using the TruSeq DNA PCR-Free Library Prep Kit (Illumina, San Diego, CA, USA) starting at the repair ends and library size selection step, with an adjusted beads ratio of 1.8 to remove small fragments. After adapter ligation, libraries were validated on a fragment analyser (Advanced Analytical Technologies).

According to the results, one or two, depending on the libraries, SPRIselect bead purifications were performed again to better select the fragments of interest. Final libraries were quantified by qPCR, normalised and pooled before 150 paired-end sequencing on the Illumina Miniseq Sequencing System with a Mid-Output Kit (Illumina, San Diego, CA, USA).

### Bioinformatics

Processing of raw sequences was conducted separately for each library using the *OBITools* package (*Boyer et al., 2016*). Initially, reads were assembled with a minimum quality score of 40. Sequences were then assigned to samples based on unique tags combinations. OTUs with less than 100 reads per library were discarded as well as those not fitting the range of metabarcode lengths. Afterwards, pairwise dissimilarities between OTUs were computed and lesser abundant ones with single nucleotide dissimilarity were clustered into the most abundant ones. We used the *sumaclust* algorithm (*Mercier, Bonin & Coissac, 2013*) to further refine the resulting clusters based on a sequence similarity of 97%. It uses the same clustering algorithm as UCLUST (*Prasad, Madhusudanan & Jaganathan, 2015*) and identifies erroneous sequences produced during amplification and sequencing. Remaining sequences were assigned to taxa using a reference database. We built a database using the *ecoPCR* for *Aves02_mix*, *Culi01* and *Plasmo01* by running an *in silico* PCR based on all the potentially amplifiable sequences available in the EMBL database (European Molecular Biology Laboratory). Since not all of these bird species were present in our database, missing sequences for frequent species occurring in the study area were generated by DNA Sanger-sequencing and added manually to the database (Data S2). Using these databases, we retained OTUs with a similarity match above 90% for Aves02_mix, 95% for Culi01 and 90% for Plasmo01. After taxonomic assignment using the abovementioned databases, all remaining OTUs were then double-checked using BLASTn on the NCBI database.

Further data cleaning and statistical analyses were performed in R 4.02. First, we used the *metabaR* package (*Zinger et al., 2021*) to assess the coverage, tag-jump rate and contamination of remaining PCR replicates. PCR replicates with too small reads count were also discarded. Removal of tag-leaked sequences was done independently for each library. This approach allowed us to discard single OTUs instead of whole PCR replicates. Remaining PCR replicates were grouped by sample and primer pair, and the mean number of sequences and mean relative read abundance (RRA) was calculated for each. We also re-calculated RRA to obtain the amount of reads per sample and primer pair relative to the total amount of sequences retrieved combining the three primers. We then visualised the differences between singleplex and multiplex in terms of primer pair, sample and taxonomy.

## RESULTS

After quality filtering, we retained 2,930,324 reads from the singleplex approach and 1,740,807 reads from the multiplex approach. We visualised the number of samples (out of 51) with a successful detection of the host-vector-parasite components in Fig. 1A, distinguishing between singleplex and multiplex assays (see Data S1 for the full dataset).

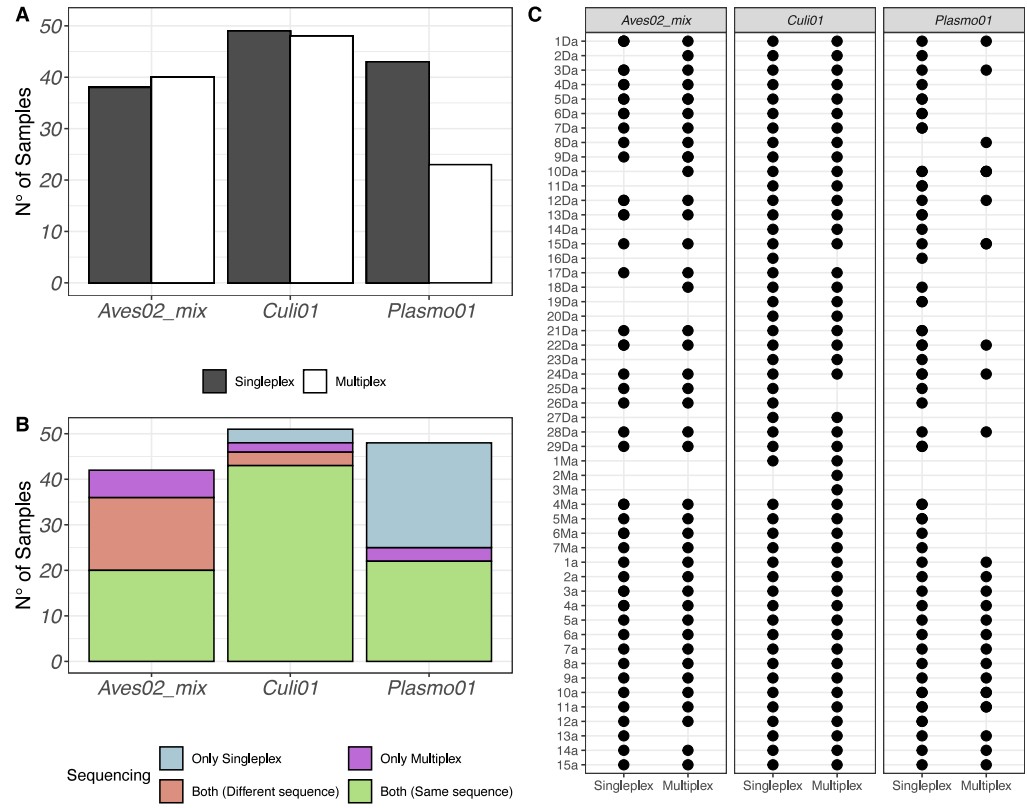

**Figure 1** (A) Overview of the detection success for each primer pair. (B) Number of samples with successful detection for each primer pair. (C) Overview of all samples and the detection success for each primer. Samples 1a to 15a correspond to mosquitoes reared in the laboratory and fed on Plasmodium-infected captive birds, which were used as positive controls.

We found a higher detection rate of *Plasmo01* in singleplex (43 out of 51) than multiplex (23 out of 51). For *Aves02_mix* and *Culi01*, the detection rates were similar between methods. Five out of 51 of the *Aves02_mix* amplifications were unsuccessful because the host was non-avian, which was verified using a generalist vertebrate primer, *Vert01* (*Riaz et al., 2011*); the putative maximum detection of *Aves02_mix* is thus 46 (Table S4).

The same results were put in perspective using the taxonomic assignment. We visualised the detections per primer pair and sequence retained (Fig. 1B). If a taxon was detected in the same sample, using the same primer pair and in both singleplex and multiplex, but the sequence detected was different, we labeled it accordingly and visualised the results in Fig. 1B. Finally, we also visualised each sample individually to compare their performance for each primer pair in Fig. 1C.

## Multiple hosts and co-infections

The DNA metabarcoding approach revealed multiple hosts, which originated from the vector feeding on various bird species, and coinfections with several *Plasmodium* lineages within a single mosquito abdomen (Fig. 2). We also detected clear differences in co-infection
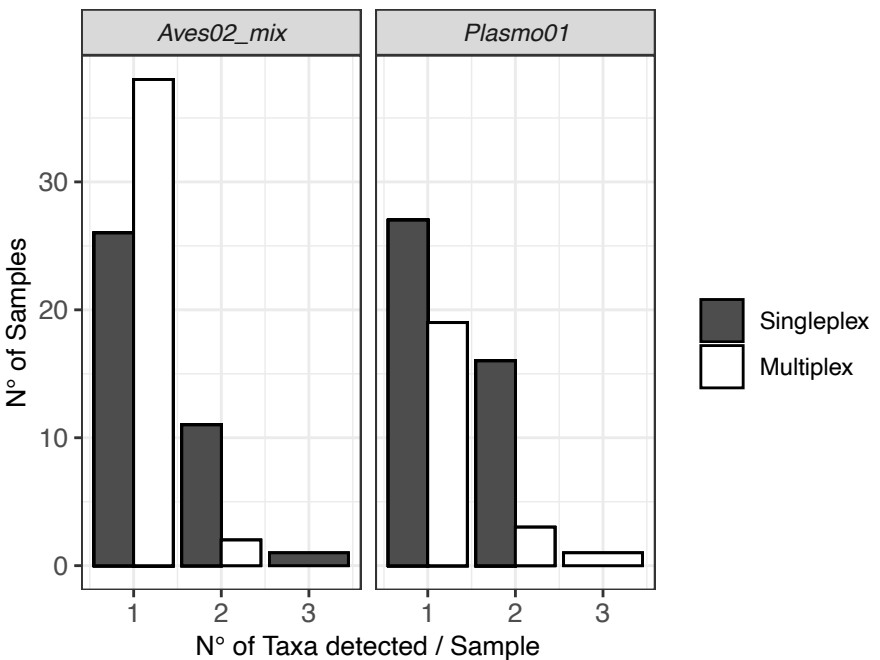

**Figure 2 Visualisation of the multiple hosts/co-infections for each primer pair, for singleplex and multiplex for all the 51 samples.** Vectors are not shown because all metabarcodes produced with Culi01 were assigned to a single taxon for each sample.

ratios between the two approaches, with singleplex being the approach with the highest number of detected co-infections (Fig. 2).

## Taxonomic detection

Based on the DNA extracted from the mosquito samples, we detected all the vectors combining singleplex and multiplex (Fig. 1). These were mainly *Culex* sp. (*C. pipiens* or *C. torrentium*, the resolution of the *Culi01* barcode being limited due to its short size), but we also detected one *Ochlerotatus* sp. mosquito. As for the hosts, we identified 12 different bird taxa, with *Parus* sp. and *Passer* sp. being the two dominant genera among the sampled mosquitoes. Together with *Passer domesticus*, *Serinus canaria* was used as positive control host. The sequences of the latter could only be assigned to family-level: *Fringillidae* (Fig. 3). Regarding the *Plasmodium* parasites, we detected four different taxa, mainly *P. relictum* and *P. vaughani*.

In order to compare the taxonomic assignment between singleplex and multiplex, we visualised the taxonomic assignment of the sequences produced with the three primer pairs (Fig. 3). For *Plasmo01* there was no discrepancy between any resulting sequence pair. There was greater variability in the sequences retrieved for *Aves02_mix* than for *Culi01*. For example, in seven samples, the sequences obtained through singleplex were assigned to *Passer domesticus* but the sequences obtained through multiplex could only be assigned to *Passer sp.*

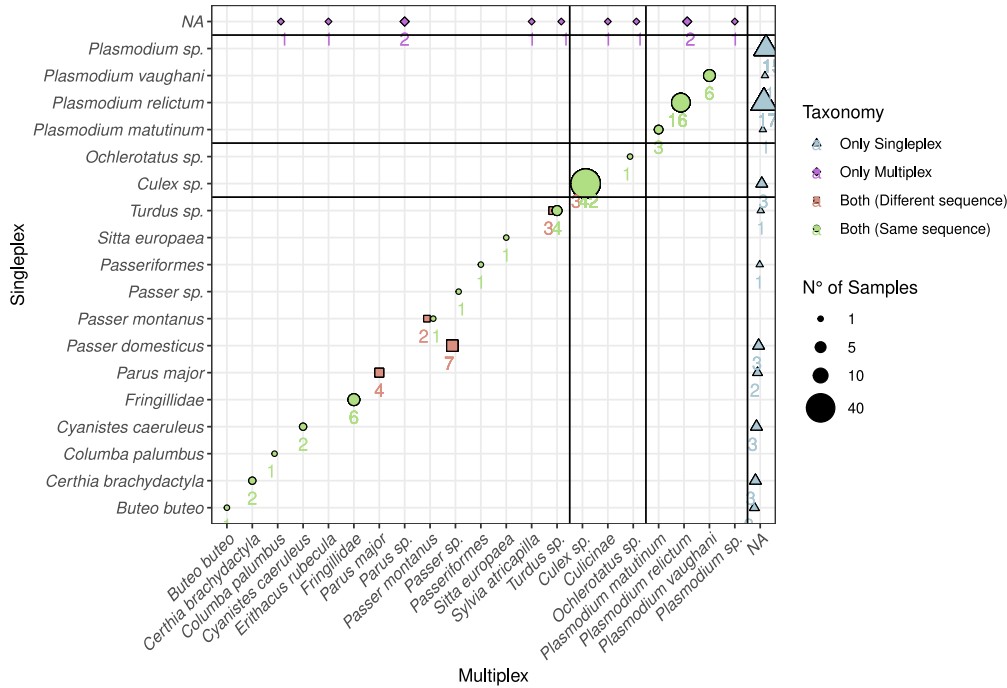

**Figure 3** **Comparison between the detected species using multiplex and singleplex and their taxonomic assignment.** The shape and colour indicate if the taxon was detected using singleplex or multiplex. If it was detected by both, we distinguish if the sequence was the same. The size of the dots and the adjacent number indicate the number of samples with identical results. The black lines separate the taxa by primer pairs and the taxa which were only detected through either singleplex or multiplex.

## Metabarcoding performance

We also explored the performance differences between the two approaches, comparing the difference in RRA for each primer pair (Fig. 4). We found the singleplex approach yielded higher RRA for *Aves02_mix*. At a lower scale, this was also the case for *Plasmo01*, although for *Plasmo01* we could not detect any taxon in many of the samples. As for *Culi01*, the relative read abundance was overall greater for multiplex than singleplex. To further visualise these differences in the performance, we compared the following variables within the two approaches: (i) the total amount of reads, (ii) RRA, (iii) RRA correcting by the primer relative concentration within multiplex (Fig. S1).

## DISCUSSION

In this study, we assessed the performance of both singleplex and multiplex approaches to study the host-vector-parasite components using the avian malaria system as a case study. The goal of this investigation was twofold: (i) to provide a proof-of-concept of the feasibility of the identification of all the components of vector-host-parasite interactions in a single PCR reaction and (ii) to highlight the methodological differences between the conventional singleplex and the multiplex approaches.

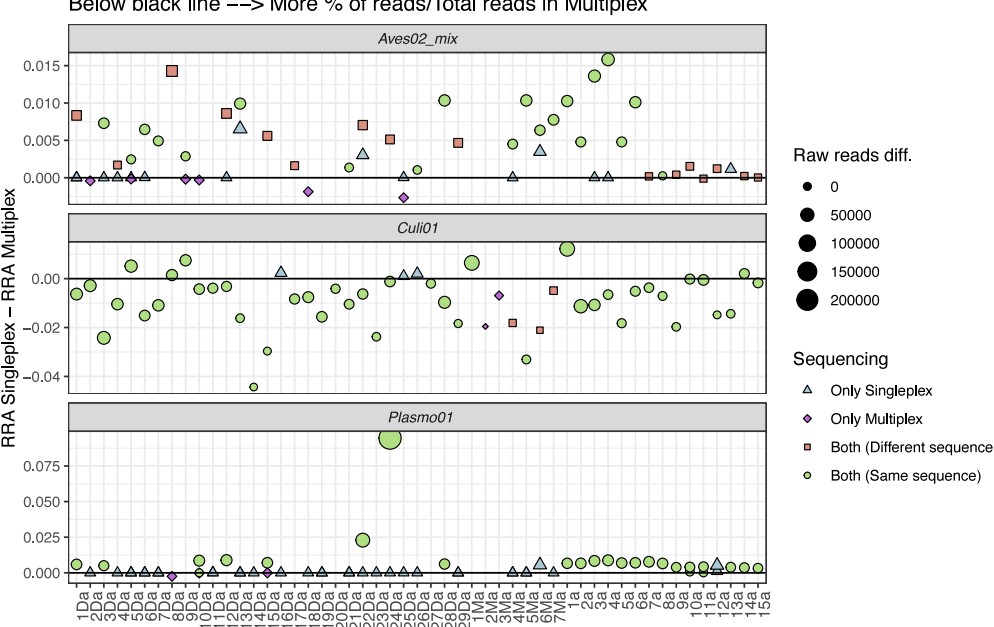

**Figure 4** **Comparison between the relative read abundance (RRA) obtained using singleplex and multiplex for each primer.** RRA was calculated as the percentage of reads assigned to each primer and sample and then divided by the total amount of reads obtained through singleplex or multiplex. The size of the dots indicates the difference in terms of absolute reads retrieved. Dots above the black line indicate that a higher RRA was found using the singleplex approach. The shape and colour indicate if the taxon was detected using singleplex or multiplex. If a sample has no dot representing a primer pair, this indicates no taxa was detected. If more than one dot represents a sample for one primer pair, this indicates that there was more than one sequence retrieved.

The use of DNA metabarcoding for detecting parasites is not new (*Diaz-Real et al., 2015*; *Ondrejicka et al., 2014*; *Šigut et al., 2017*), but this approach had until recently been limited by the separate detection of each component of a vector-borne system. Despite the great accuracy of a singleplex approach, a better comprehension of the interactions between the components of the host-vector-parasite triad might enable a better understanding of their evolutionary dynamics and improve prevention campaigns of vector-borne diseases. A multiplex approach using the vector as a template (*e.g.*, blood meals) could highlight which host species are mostly infected or which are the dominant associations between certain parasite, vector and host species. These are just a few examples of potential applications for elucidating the interactions between the components of a vector-borne system using multiplex assays (see also *Campana et al. (2016)* for an in-solution capture approach, and *Orantes et al. (2018)* for a RADseq-based pipeline).

In terms of detection of each component of the tripartite association, our results reveal clear differences when comparing singleplex and multiplex. The results were similar for both host and vector, but the detection of the parasite was higher using the singleplex approach (Fig. 1A). We suggest this is due to the low initial DNA concentration of *Plasmodium* in the multiplex mix. The parasite template DNA competes with the DNAs

from host and vector to be amplified in the multiplex, and its low initial concentration (in relation to very low infection intensities in the blood of hosts—chronic stage of infection, *Pigeault et al., 2018*) translates into lower amplification, and thus lower detection. Host and vector DNA is more abundant in the blood-meal DNA extracts, and given their shared resources for amplification, have a greater likelihood to be amplified. This is also the reasoning behind the standard use of a nested PCR protocol developed more than 20 years ago to detect and identify haemosporidian infections (*Bensch et al., 2000*; *Waldenström et al., 2004*). In this case, two PCRs are necessary: the first one amplifies a longer fragment of the mtDNA cytochrome *b* gene (approx. 580 bp including primers) which contains the barcode of the target parasite; the second one amplifies a smaller fragment (approx. 520 bp) using the first amplicon as template. This method allows for an increased sensitivity and specificity because two primer sets successively bind to the same target sequence, although nested PCRs increase the risk of PCR mutations and cross-contaminations. However, amplification of DNA fragments of this length can be difficult with potentially degraded samples such as blood meals in the midgut of the vector (*Jo et al., 2017*). When comparing the sequences retrieved using both approaches, we observed clear differences among primer pairs. *Plasmo01* and *Culi01* yielded the same DNA sequence in both approaches, whereas half of the samples amplified with *Aves02_mix* produced different sequences (Fig. 1B). This was expected given that *Aves02_mix* is a mix of three primer pairs. We suggest that the competition between these three primer pairs with the *Culi01* and *Plasmo01* in multiplex could have bound unevenly to the template DNA producing slightly dissimilar sequences, but we cannot rule out potential heteroplasmy. However, the taxonomic assignment of these sequences revealed these were equally resolutive (Fig. 3), except for one case. This confirms the equal performance of both approaches to retrieve consistent taxonomic information, despite the small differences within each sequence.

Regarding co-infections, they were more frequently detected when using singleplex. These results highlight the higher resolution of the data when a single PCR reaction is dedicated to each primer pair, despite the higher cost and greater time investment when analysing a large number of samples compared with multiplexing. When designing the laboratory procedure for an experiment, using one approach or another has to be assessed having this trade-off in mind.

To further understand the performance differences, we compared singleplex and multiplex in terms of the RRA for each approach (*i.e.,* singleplex or multiplex), primer and sample, calculated dividing by the total amount of reads of each approach as reference, instead of the sum of reads per sample (Fig. 4). We did this to account for the amplification and sequencing biases across primer pairs. The mix of primers for the multiplex approach was not equimolar; we purposely increased the concentration of the *Plasmo01* and *Aves02_mix* pairs. This was done in order to reduce the potential over-amplification of *Culi01*, given the blood-meals were extracted from mosquito abdomens, and we expected their DNA to be more abundant and of better quality. We used the singleplex results as unbiased reference of primer performance for each sample, which highlighted the differences between primer pair performance when used together in the multiplex. The results were distinct for each primer pair: RRA was greater in singleplex for *Aves02_mix* and

*Plasmo01* (although by a smaller margin); and greater in multiplex for *Culi01*. We interpret these results as follows: the reduction of *Culi01* concentration did not compensate for the abundance and quality of mosquito DNA, which still represented a higher proportion of reads. This can be observed when comparing Fig. S1: correcting for the primer concentration in multiplex only reduced *Culi01* amplification, but not *Aves02_mix* or *Plasmo01*.

Nevertheless, this did not affect the detection of avian hosts (only three samples that were amplified in singleplex were not amplified in multiplex). However, it had a great impact on the *Plasmo01* detection, as half of the *Plasmodium* detections occurred only through singleplex. This suggests the low abundance of *Plasmodium* DNA in the blood-meal was not compensated by the greater concentration of *Plasmo01* and undermined the detection potential of the multiplex approach. However, in terms of *Plasmo01* RRA, we did not find clear differences between the two approaches (Fig. 4).

Put in perspective, the use of multiplex offers major advantages in terms of cost and time over the use of singleplex. Their detection and taxonomic performance are comparable for both host and vector. However, the parasite, and its low initial DNA concentration, poses greater challenges. We suggest the relative concentration of parasite primers should be greatly increased compared to the host and vector ones in order to increase the reliability of the detection results when using a multiplex approach. More generally, several relative primer concentrations within a multiplex could be tested, in order to optimise detection of each of the targeted taxa involved in the final assay. This can be for instance achieved by using a fragment analyzer, which quantifies DNA size and relative abundance, provided that the barcode lengths have non-overlapping ranges between the three components. Doing so, the relative primer concentration above which the parasite (or least abundant component) is revealed in the multiplex can be adjusted to maximize its detection. New generation *Taq* polymerases optimized for co-cycling of different PCR targets (*i.e.,* multiplexing) can also be used in the assay (*e.g.*, Platinum II *Taq* Hot-Start DNA polymerase, Thermo Fisher), improving the likelihood to reveal the minor component in a DNA mixture after the amplification step. Furthermore, other combinations of DNA primers can be developed in the multiplex assay, either by targeting other taxonomic groups or by achieving different taxonomic coverage and resolution, in the case of the malaria system but also in any vector-borne interaction involving hematophagous arthropods.

Overall, we conclude from this proof-of-concept study that the applications of multiplex are promising, but further research is needed to better understand multiplex detection probabilities and to refine the technique for large-scale applications in vector-borne disease prevention management.

## ACKNOWLEDGEMENTS

We thank M Baur, N Remollino, C Stoffel and J Wassef for their help in the laboratory.

### Funding
This work was supported by a fellowship in Life Sciences (Faculty of Biology and Medicine) from the University of Lausanne to Eduard Mas-Carrió, and by a Swiss National Science Foundation grant (nr. 31003A_179378). The funders had no role in study design, data collection and analysis, decision to publish, or preparation of the manuscript.

### Grant Disclosures
The following grant information was disclosed by the authors:
Life Sciences (Faculty of Biology and Medicine) from the University of Lausanne.
A Swiss National Science Foundation grant: 31003A_179378.

### Competing Interests
The authors declare there are no competing interests.

### Author Contributions

- Eduard Mas-Carrió analyzed the data, prepared figures and/or tables, authored or reviewed drafts of the article, and approved the final draft.
- Judith Schneider performed the experiments, authored or reviewed drafts of the article, and approved the final draft.
- Victor Othenin-Girard performed the experiments, authored or reviewed drafts of the article, and approved the final draft.
- Romain Pigeault performed the experiments, authored or reviewed drafts of the article, and approved the final draft.
- Pierre Taberlet analyzed the data, authored or reviewed drafts of the article, and approved the final draft.
- Philippe Christe conceived and designed the experiments, authored or reviewed drafts of the article, and approved the final draft.
- Olivier Glaizot conceived and designed the experiments, performed the experiments, authored or reviewed drafts of the article, and approved the final draft.
- Luca Fumagalli conceived and designed the experiments, authored or reviewed drafts of the article, and approved the final draft.

### Field Study Permissions
The following information was supplied relating to field study approvals (*i.e.*, approving body and any reference numbers):

Field experiments were approved by the Direction générale de l'environnement (DGE), Section biodiversité et paysage, CH-1014 Lausanne (Switzerland). Approval number: 2019-3978.

### Data Availability
The DNA metabarcoding data generated for this study is available at Dryad: Mas-Carrió, Eduard; Schneider, Judith; Othenin-Girard, Victor et al., 2024. Data from:

Multiplex *vs*. singleplex assay for the simultaneous identification of the three components of avian malaria vector-borne disease by DNA metabarcoding [Dataset]. Dryad. https://doi.org/10.5061/dryad.ht76hdrqg.

## Supplemental Information

Supplemental information for this article can be found online at http://dx.doi.org/10.7717/peerj.19107#supplemental-information.

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
