# Peer review of "Multiplex vs. singleplex assay for the simultaneous identification of the three components of avian malaria vector-borne disease by DNA metabarcoding"

_PeerJ, doi:10.7717/peerj.19107_

## Round 0.1 · original submission · Major Revisions

The authors of this paper evaluated a new multiplexing protocol designed to simultaneously detect vertebrate hosts, Plasmodium parasites, and mosquito vectors in a single reaction. This approach has the potential to uncover hidden mixed infections that are often missed by single amplicon PCR with Sanger sequencing. When comparing their results with singleplex PCR analyzed through Illumina sequencing, they found similar detection efficiency for vertebrates and insects, but lower efficiency for Plasmodium parasites in the multiplex approach. Therefore, this manuscript addresses an essential issue in the field. Two expert reviewers have thoroughly examined this manuscript and provided feedback to the authors for improving their study. The reviewers have raised important and valid criticisms regarding the new methodology proposed. I agree with them and emphasize that the authors must respond to the raised points with the utmost precision and make all necessary revisions in the manuscript before submission.

Reviewer 1 ·

Basic reporting

In this study Mas-Cario and colleagues, test the efficiency of a new multiplexing protocol to detect vertebrate host, Plasmodium parasite and mosquito vector species in on reaction, thereby also potentially revealing hidden mixed infections which often remain hidden when performing single amplicon PCRs combined with Sanger sequencing. The author compare their results with singleplex PCR reactions through Illumina sequencing and find that the detection efficiency is similar for vertebrates and insects but is lower in multiplex compared to singleplex for Plasmodium parasites.

The study is generally valuable, makes many good points and undertakes development in a useful and needed direction but also has some shortcomings and loose ends, which should be addressed more directly and with suggestions for how to improve in the discussions.
Firstly, given the established protocols in the field, which are general for different haemosporidian genera, a Plasmodium-specific assay appears rather as a step back. Is it really impossible to design suitable primers for all haemosporidians? The authors might argue that for mosquitoes only Plasmodium primers make sense but this is how information about abortive infections, when they abort and developments in untypical vectors may be missed.

Secondly, the used amplicon for Plasmodium barcoding is really small at 44 bp. The authors argue that this is useful due to DNA degradation during digestion in the vectors. Some numerical support for this otherwise logical effect would be needed. Also, what is the resolution of lineages given this small fragment? I guess some common lineages would remain practically indistinguishable. In general, sequencing 44bp out of 250 bp possible seems rather inefficient. I would discuss rather accepting a gap of in the middle of the sequenced amplicon.

Thirdly, given the found lower efficiency of the multiplex to detect mixed infections compared to the singleplex, the authors suggest adjusting the primer concentrations. This is really a minor tweak and I am surprised that the authors did not perform it themselves in order to present a more fulfilled and finally usable protocol. It sounds more like a half-baked protocol is being presented currently. Possibly the study results from a time- and resource-limited student project which would explain the truncated development. This is fair but still the relative improvement of efficiency for the smallest amplicon could be tested and visualized at least on a gel or quantified through qPCR (for which however the short size of the amplicon might be a problem). At least a discussion of how much work this might be and why the authors have not tried it would be appropriate. Currently, the discussion rather implies that primer concentrations can be varied by 10 nM and a full MiSeq run and corresponding publication may be performed for every iteration, which would be highly redundant.

Some more specific comments:

L68 some studies also score all three e.g. Dumonteil, E., Ramirez-Sierra, M.-J., Pérez-Carrillo, S., Teh-Poot, C., Herrera, C., Gourbière, S., & Waleckx, E. (2018). with Illumina, and Chakarov, Wiegmann, Kampen, Werner, Bensch (2020) with Sanger.

L138, 159 Please specify here that you mean amplicon/product DNA and not bands of total genomic DNA, whose size can also be envisioned and bands there expected.

L153 While not strictly incorrect, I would advise to give the used Plasmodium primers a different name since Plas1 is already a name of popular primers used for general haemosporidian detection designed by Pérez-Rodríguez, de la Puente, Onrubia & Pérez-Tris, 2013. The similarity of primer naming can lead to misunderstandings in labs using diverse methods and protocols.

L198 If I understand correctly the Plasmodium amplicon should be quite a small fragment. Is it possible it was removed by the beads?

L278 could not

L297 – how are blood meals the vector component? This is either wrongly connected or needs to be explained more.

L305 - low initial relative DNA concentration of Plasmodium? The actual problem is not that is low but that it is competing with other reaction for the polymerase

L315 these nested PCRs are not directed only at Plasmodium, are they?

L320 is there any literature support that 570 bp is more problematic than shorter fragments? 570 is already quite short in chromosomal terms.

L330 It would be better also to include some discussion if this approach can work with a greater variety of hosts and corresponding richer haemosporidian fauna. The signal there might be too mixed.

L359 I would have liked to see a more critical discussion of the size of the chosen fragment to characterize the parasites. This seems to be to small to really disentangle more complex mixes of parasites. Also a discussion of the drawback of amplifying only Plasmodium instead of all haemosporidians is needed. While mosquitos are mainly vectors of Plasmodium, it is valuable information which should be revealed through such analyses all parasites which about development in these vectors and their retention times.

L369 – this semi-final conclusion sort of suggests that in the style of this manuscript every tweak of concentrations of primers in the mix should lead to a manuscript like this one. Why did the authors refrain from testing this suggestion? I imagine costs were a substantial factor (and/or the fulfilment of requirements for a corresponding thesis) but this could possibly be done even with methods cheaper than MiSeq, such as qpcr?

Experimental design

A good start. Improvements should be discussed more specifically.

Validity of the findings

Good but could easily become more general.

Reviewer 2 ·

Basic reporting

The manuscript examines the possibilities of the multiplex PCR to investigate samples (collected vectors) faster and simpler than using singleplex PCR.
I understood that the newly developed multiplex PCR can be succesfully used for the detection of vector and host, but it is more complicated with parasite detection, as the concentration of parasite DNA, because it is lower comparing with the concentration of DNA of hosts and vectors. That is why a nested PCR is used very often for the detection of parasites.

Only Plasmodium parasites have been investigated, but birds and mosquitoes can be infected not only by Plasmodium parasites, but also by other apicomplexan parasites (Haemoproteus) and this issue should be also discussed in the paper.

I would like to see a clear explanation of how the coinfections and the presence of different hosts in the same mosquito were determined.

Experimental design

Line 113:
36 wild gravid female mosquitoes were captured…

It should be described if the blood inside abdomen of mosquitoes was visible, or not (it is known that in case the blood is not visible it is almost impossible to determine the host). The detection of blood source inside the mosquito is complicated due to the digestion of blood over the time. It is also strange that almost all collected mosquitoes (see Fig. 1c) were found to be infected with Plasmodium parasites, as the prevalence of Plasmodium in wild caught mosquitoes is known to be up to 30%, but usually it is lower than 10 %.

Validity of the findings

No comment

Additional comments

It would be good not to use „blood sample“ (see line 525), as mosquito abdomen were investigated, not blood samples.

Line 332
Regarding co-infections, they were more frequent when using singleplex.

They were not more frequent, but they were not frequently detected….

Fig. 2
That is shown for the “3 (No of Taxa detected)”? Did all Aves02mix were detected using singleplex?

I don't understand why, when citing references in the text, sometimes spaces are left between words, but usually they are not. What do the notes e.g. mean? (e.g. (Kocher et al., 2017)))?

---

## Round 0.2 · Minor Revisions

The authors addressed the main concerns of the reviews. However, the revised manuscript still deserves attention. Please provide point-to-point responses according to the comments made by Reviewer #1 in the new version of your manuscript.

Reviewer 1 ·

Basic reporting

Sufficient.

Experimental design

Somewhat premature and partly flawed. Should be introduced through constraints and discussed correspondingly.

Validity of the findings

Intrinsically consistent but of limited value for general use. Usable as an intermediate step for creating and improving similar protocols.

Additional comments

The manuscript is of interest but still needs substantial improvements, some of which were already asked for in the previous review round but have been ignored (e.g. renaming Plas01, more nuance in the introduction and discussion). In the first parts of the manuscript the claims are too bold for what is presented in the end. This is obviously a proof of concept study, rather than a final usable multiplex assay. This is stated at the beginning of the discussion but not in the introduction where it also belongs. It should also be flagged as such already in the abstracts and handled accordingly already in the introduction. And posed as an open question - can we improve compared to the singleplex and how much? No statistics are used to test the differences between both but might be adequate. The concept study should not go to waste and should be published, but it also builds an introduction by discussing technical issues and implying solvability, but it finally does not solve them in its current state. So, using a more nuanced language and open outcome prerogative is necessary. Also the usefulness of the limited phylogenetic extent is completely ignored and should be discussed at least a bit. In the end such limitations might speak against multiplexing, quite probably against this specific protocol and this would be the wrong take-away message. The message should be - this is a first attempt and more developments in this direction are needed.

Specific comments:

L38 - We found both to have similar performances in terms of detection for the host and the vector, but singleplex performed better than multiplex for the parasite component. – This is not consistent with the content of the manuscript true. The singleplex seems to substantially outperform the presented multiplex. Thus, the new presented assay does not perform “Accurate detection and identification of vector-host-parasite systems” as the beginning of the abstract suggests.

L42 - Overall, the results show that multiplex DNA metabarcoding is an effective approach – can be an effective approach – currently it is not very effective. It misses some parts of the data and patterns.

L43 - Our insights will not only refine laboratory procedures – can refine. Developed at the level at which they are, they would lead to random or non-random subsampling of patterns occurring in nature and are though to be elucidated with these analyses.

L43 - Our insights will not only refine laboratory procedures, but also enhance research efforts and medical diagnosis of vector-borne diseases, the spread of which is globally exacerbated by current climate change. – The whole sentence is inappropriately self-glorifying given the limitations of the presented procedure and results. I propose including may and can in the first two subsentences here. And generally the “proof of concept” accent and wording.

L71 -Some studies have already used singleplex to identify the host-vector-parasite triad, e.g. using blackflies (Dumonteil et al., 2018) or triatomines (Chakarov et al., 2020). – references here seem mixed up.

L76 - Currently, the application of multiplexing is however limited, because the different primer sets used need to have a similar annealing temperature to correctly bind to the target region while keeping the amplified barcodes both taxonomically informative and within the sequencing thresholds (up to 300bp, MiSeq Illumina high-troughput sequencing). ). This limitation has constrained the application of multiplexing, but… – This argument is deprecated and unnecessary. The pseudo-problem can be solved with sufficient play with primer design and MiSeq. Miseq can do 2x 300 bp. Please remove this sentence. The factors for not using multiplexing and Illumina are quite different, such as simplicity and tradition with Sanger analyses, bioinformatic reservations by a limited untrained crowd, general lack of multiplexing routine among in the concerned community, downward scalability – few people of the community need or can invest in the number of samples covered by an even small Illumina run for one project.

L132 – why avian – Mosquitoes could have sucked on a spectrum of other vertebrates. These can consequently not be identified as potential targets of the tested mosquitoes. While it is clear that the assay is designed specifically for avian malaria studies, most future users will likely not like to lose information of potential non-avian hosts.

L132 - The presence of amplicons at the expected molecular size were verified on a 2% agarose gel
This should be “The presence was verified…”

L132-141 This seems like a very labourous and inefficient way to substitute the universal vertebrate barcoding primers , which should be fully within the 2x300 bp MiSeq range. Why were not all samples treated as in L142-143?

L151 – I reiterate – The name of the primer should be changed! In their rebuttal the authors write “We used Plas01 following the same pattern of taxonomy-target + number as for all the other primer pairs used in our lab and between our collaborators. Since we provide the primer sequence explicitly, we consider it is adequate to maintain the name and the risk of confusion is very low.” – This argument is deprecated. You are not publishing the primer for your own private but for public use. If the sequence is the primer identifier, they would not get any names. Any reasonable lab dealing with barcoding would have all sets of adequate barcoding primers – it is very counterproducitive therefore to have two different primers called Plas01 in the boxes of one lab which can be distinguished only by their sequences. This is bound to lead to mix-ups! As in the case of haemosporidian lineages for the MalAvi database, naming the primer which comes second should be done with an alternative name/number, i.e Plas02. It should be quite easy to rename it for use within your lab, especially since it will likely not be upgraded and replaced in the near future.

L151 – this 57% species determination and no lineage level – this can not be qualified as “high taxonomic resolution”. In the previous response the authors write that the primers achieve “100% resolution at genus and family levels” which is tautological given that they have been specifically designed for and tested within the genus Plasmodium. Please present another relevant argument why these primers were chosen.

L156 – GenBank has orders of magnitudes more haemosporidian cytb Fragments than 12, many more. The more adequate database for this would anyway be MalAvi. The authors possibly refer to complete mitochondrial genomes – given that the target region is in cyt b there is no reason to use only these. I.e. this argument is unconvincing and needs to be substituted. It is ok to say that you used a limited set of cyt b sequences to construct a consensus primer sequence.

L175 – given that your argumentation why multiplexes are not used was disparate primer annealing temperatures (L77), now presenting primers which have different ones and are optimally tested in singleplex at different annealing temperatures is inconsequent and possibly not a great solution to the posed problem. 5 degrees difference might not be a problem but the singleplexes maybe should have been all optimally tested at 55. You can discuss the pros and cons of this incomplete factorial comparison.

L202 so the whole assay was designed so it could fit to the Miniseq reads output! It is ok to put this as an objective constraint, but in this context it is important to point out this is a proof of concept case study, rather than a protocol to be used by the masses for decades such as the 16S metabarcoding.

L254 detected co-infections.

L260 is this not the only purpose of this barcode?

L281 – Fig. S1 only has 3 rows. There is no (iv) normalised RRA. B and C are practically identical. This does not seem correct, but if it is correcting for relative primer concentrations does not lead to changes and can be abandoned, block C can be deleted. The whole figure is very difficult to understand.

L285 – This „proof-of-concept” claim and phrasing are adequate and should also be present in the Abstract.

L294 – this is a might, not a would. I wouldn’t know of a single study who have kept this promise of informing prevention campaign for novel diseases, so it remains a hypothetical.

L295-L300 overall the multiplex approach makes a promise of cost-efficiency. Otherwise, many labs and studies score the separate components in parallel as singleplexes.

L317 they increase the risk of PCR mutations. The cross-contamination frequency should be pretty much the same as in any other PCR.

The Discussion is the most balanced part of the manuscript.

---

## Round 0.3 · accepted · Accept

The authors have addressed all the reviewer's comments appropriately. Therefore, the manuscript is now ready for publication.

Reviewer 1 ·

Basic reporting

The manuscript now reeds very nicely.

Experimental design

Sound.

Validity of the findings

Provide a basis for further development.